# Building Rehabilitation: A Sustainable Strategy for the Preservation of the Built Environment

Ana Martha Carneiro Pires de Oliveira [1],*, João Carlos Gonçalves Lanzinha [1] and Andrea Parisi Kern [2]

1   C-Made—Centre for Building Materials and Technologies, Department of Civil Engineering and Architecture, University of Beira Interior, 6201-001 Covilha, Portugal; jcgl@ubi.pt
2   Graduate Program in Civil Engineering, University of Vale do Rio dos Sinos, Sao Leopoldo 93022-750, Brazil; apkern@unisinos.br
*   Correspondence: anamarthaeng@gmail.com

**Abstract:** Building rehabilitation and sustainability must go hand in hand to ensure the preservation of the built environment and environmentally conscious practices. Construction is one of the most polluting industries with a high impact on the carbon footprint. Thus, building rehabilitation appears as an effective strategy to reduce this impact, promoting the reuse of more efficient materials and technologies. This study focuses on the rehabilitation of existing buildings as a sustainable strategy and presents the quantitative profile of academic publications in the last 10 years, showing the main themes studied. The results of the sample surveyed on the Scopus platform show an increasing number of publications in the period surveyed (2012–2023), which shows a growing academic interest in the topic. It is possible to see that the publication trend line is ascending and that the largest number of articles investigates building rehabilitation, followed by the rehabilitation of school buildings, energy issues, rehabilitation methods, materials technology, water issues, and sustainability related to rehabilitation.

**Keywords:** sustainable development; rehabilitation; construction

## 1. Introduction

The construction industry is a sector of the world economy that contributes to the pollution of the planet in terms of $CO_2$ (carbon dioxide) emissions, equivalent to 39% of global emissions. This industry is an important economic sector, and each country analyzes its ability for economic growth with the smooth functioning of the construction industry. Khaertdinova [1] discusses the recent development of the construction industry, a sector that, despite showing slow development in the last two decades, continues to use the same polluting technologies of the past. For this reason, the authors say, there is a need for innovation in building materials, but often to the detriment of the environment. European countries are looking for sustainable construction methods, such as the principles of reusing materials during the deconstruction and rehabilitation of buildings.

Brazil, Portugal, and other countries associate rehabilitation with the energy efficiency of a building. Azevedo and Tavares [2] point out that this association is not wrong. However, it is outdated because the concept and application of rehabilitation in real estate projects are changing concerning the importance of energy efficiency in buildings. Gupta and Chakraborty [3] point to the need for more energy-efficient building designs to address the rising cost of energy supply in addition to the global energy crisis. Kats [4] discusses the demand for more public policies and investments in innovative technologies that increase the energy efficiency of buildings. Mastouri [5] emphasizes the importance of existing regulatory mechanisms concerning building performance requirements and the need to integrate passive local and bioclimatic approaches to improve energy efficiency in buildings and reduce energy consumption while considering environmental concerns.

Rahman and Ali [6] claim that construction activities contribute to various forms of pollution, including air pollution, water pollution, noise pollution, and solid waste pollution. Manzhilevskaya [7] specifically focuses on air pollution from fine dust from construction sites and suggests protective measures to reduce these emissions.

This literature review focuses on sustainable strategies for the rehabilitation of existing buildings and supplies a quantitative analysis of scholarly publications spanning the last decade. It aims to show the predominant themes studied in this context. The analysis of the data obtained from the Scopus platform reveals a notable increase in publications during the investigated period (2012–2023). This upward trend signifies a growing scholarly interest in the subject.

## 2. Literature Review: Publications on Rehabilitation Aiming at Sustainability in the Last 10 Years

There are different methodologies for conducting literature reviews. Gómez-Luna [8] presents a method for finding, organizing, and analyzing information in any field of research. Laghrabli [9] introduces a new framework for literature review analysis using association rule mining. Torres-Carrión [10] proposes a method for a systematic review of the literature applied specifically to engineering and education. Costa and Zoltowski [11] focus on the methodology for effective bibliographic research, with delimitation of the question to be researched and the choice of data sources, including the choice of keywords for the search, which is achieved through the organization of a spreadsheet that performs the selection before subsequent reading of the selected material.

For this work, the proposal of a literature review in search of the state of the art for the given subject followed the approach of Costa and Zoltowski [11]. To this end, this research followed some steps presented by the authors:

1.  Delimitation of the question to be researched.
2.  Choice of data sources.
3.  Selection of keywords for the search.
4.  Search and storage of results.
5.  Selection of articles by abstract, according to inclusion and exclusion criteria.
6.  Extraction of data from selected articles.
7.  Evaluation of articles.
8.  Summary and interpretation of data ([11], p. 54).

The delimitation of the issue to be researched is, as already presented, the rehabilitation of buildings and sustainability; however, according to Costa and Zoltowski [11], this delimitation is still quite comprehensive and allows for many variables, so the search was carried out in only one database, the Scopus platform, in which the question was presented. The search keywords, in English, were, after many attempts, the words (a) refurbishment and (b) building retrofit, which resulted in some doctoral theses and master's dissertations, as well as scientific articles.

After finding the search results, the texts were saved, and a list was created in a data spreadsheet to filter the data and organize the results to present research in rehabilitation and sustainability in the field of buildings. The scientific articles found using the keyword refurbishment presented, after delimiting the theme and the area of civil engineering, 452 articles related to the theme of rehabilitation. These articles were published between 2012 and 2023 (Table 1).

It is possible to see that the trend line is ascending until 2021, although in the years 2018 and 2020 the number of publications was lower than the average of previous years, with a drop in publications in 2022 and 2023 (a year that is still ongoing). Table 2 presents the central themes of the building rehabilitation studies in the sample of articles.

**Table 1.** Publications on building rehabilitation from 2012 to 2023.

| Year of Publicationation | |
|---|---|
| 2012 | 24 |
| 2013 | 31 |
| 2014 | 44 |
| 2015 | 51 |
| 2016 | 57 |
| 2017 | 54 |
| 2018 | 37 |
| 2019 | 42 |
| 2020 | 33 |
| 2021 | 49 |
| 2022 | 24 |
| 2023 | 6 |

**Table 2.** Investigated themes of building rehabilitation.

| Category | |
|---|---|
| Energy/Water | 178 |
| Materials | 38 |
| Buildings | 35 |
| Assessment/Inspection | 34 |
| Sustainability | 35 |
| Costs | 31 |
| Technology | 27 |
| Heritage | 18 |
| Health | 16 |
| Schools | 10 |
| Safety | 7 |
| Accessibility | 2 |

According to Table 2, within the sample, there are a substantial number of studies on energy/water, a major concern of developed countries. This theme is related to energy efficiency, which is one of the recurring themes when talking about sustainability today. The second most addressed topic, which is also associated with sustainability, is the choice and use of materials that are more suitable for civil construction.

Examples of references on each topic are presented below:

1. Energy/Water—"Method of Planning Repairs of the Installation including Building Waste" [12] is a work that addresses the authors' concern with rehabilitation projects that disregard the need for adequacy and/or repairs of the hydraulic and sanitary sewage systems of buildings, which, according to the text, leads to more future expenses when, after rehabilitation, the new loads of use incur an increase in pressure on the non-renewed hydraulic installations.

2. Materials—"Aerogel materials for heritage buildings: Materials, properties and case studies" [13] is an exemplary work on materials applied to rehabilitation. The study in question points out that there are few studies about the results of the application of aerogel, a thermal insulation material applied in rehabilitation interventions. The

text presents a few case studies with the use of aerogel in historic buildings with satisfactory results on the improvement of thermal comfort and energy efficiency.

3.  Buildings—The survey recovered works that fit into the category 'building', which deals with works related to interventions in buildings of the most varied origins. "Constructive Retrofit Guidelines for Social Housing Buildings in Beira Interior Region, Portugal, for Actual and Future Climate Scenarios" [14] presents a reflection on rehabilitation focused on thermal comfort in the face of new climate scenarios in which there is an increase in heat waves during the summer, as well as colder winters in many regions of the world. The case study was carried out in the city of Covilhã, Portugal, which has very different seasons.

4.  Assessment/Inspection—The evaluation methodologies for rehabilitation are procedures that help engineers in decision making and in suggesting the interventions to be carried out, where in many cases they are focused on the envelope and rarely focus on one of the aspects considered most important in developed countries, which is energy efficiency. Hence, the article "A decision support system for scenario analysis in energy refurbishment of residential buildings" [15] is different in that it proposes an evaluation methodology for rehabilitation focused on electricity savings.

5.  Sustainability—Rehabilitation has as its main concern the sustainability of the construction and the constant search for materials and solutions that will help reduce carbon emissions and reduce waste from the intervention works. "Mitigating Climate Change in the Cultural Built Heritage Sector" [16] is a text that addresses sustainable climate and energy issues in old buildings.

6.  Costs—Cost analysis is a useful tool to stimulate the renovation of existing buildings by highlighting the minimum overall cost that private or public developers must bear to achieve appreciable energy savings over the life of the building. The article "Social housing refurbishment in Mediterranean climate: Cost-optimal analysis towards the n-ZEB target" [17] performs a series of analyses using the same energy performance simulation tool, highlighting the deviations between the best solutions and the minimum interventions to lower energy consumption.

7.  Technology—The search for technologies that help with rehabilitation issues is the focus of the text "Modelling and Simulation of Building Integrated Concentrating Photovoltaic/Thermal Glazing (CoPVTG) Systems: Comprehensive Energy and Economic Analysis" [18], in which a new photovoltaic energy harvesting technology is presented.

8.  Heritage—Historic buildings are important because they reveal the history of a country or region. They are also tourist landmarks that identify districts to explore. Their rehabilitation and conservation must be undertaken to support and preserve historical heritage. This is the case presented by the article "A review on the technical performance criteria of post-occupancy evaluation (POE) of refurbished heritage museum buildings" [19], which uses the post-occupancy evaluation (POE) methodology as a goal to evaluate the performance of the building after undergoing a rehabilitation process.

9.  Health—Air quality, which includes temperature variations and humidity, is a problem in buildings, especially in historic buildings. Air quality interferes both with the preservation of cultural assets and with the health of the people who circulate inside these buildings. The article "Numerical multi-physical approach for the assessment of coupled heat and moisture transfer combined with people movements in historical buildings" [20] analyzes three parameters over time to understand how the control of indoor microclimatic conditions works. The authors develop a transient simulation model for heat and humidity transfer considering people's movements.

10.  Schools—"From Indicators to Strategies: Key Performance Strategies for Sustainable Energy Use in Portuguese School Buildings" [21] is an article that studies energy efficiency in schools in Portugal that have undergone rehabilitation processes.

11.  Safety—"Maintenance, reconstruction, and Prevention for the Regeneration of Historic Towns and Centers" [22] is an article in which the central concern is with the

maintenance of rehabilitated historic buildings in terms of their ability to withstand earthquakes.

12.  Accessibility—An article that also addresses schools discusses accessibility as an evaluation factor in relation to the rehabilitation strategies to be conducted. The text "Early Decision-making for School Building Renovation" [23] attests that the discussion of accessibility is remarkably close to the rehabilitation of school buildings.

The bibliographic survey conducted to reflect on the sustainability of building rehabilitation proved to be especially important in proving that, although rehabilitation is directly associated with the field of civil engineering and architecture, it also has many elements of sustainability. As sustainability is an emerging issue in different areas of knowledge and human industrial activities, it is important to highlight that the rehabilitation of buildings is also a sustainability action since rehabilitation has at its core the preservation of heritage and the environment.

## 3. The Concept of Sustainability in Civil Construction

The concept of sustainability appeared with the creation of the World Commission on Environment and Development (WCCD) by the United Nations General Assembly in 1983 [24]. In 1987, the report "Our Common Future", or the "Brundtland Report", described, for the first time, the concept of sustainable development [25,26]. In 1992, the United Nations Conference on the Environment (ECO 92), held in Brazil, debated and updated the concept to expand environmental preservation and ensure a better quality of life for all humanity [27].

Chakravarty [28] says that sustainable development transcends environmental sustainability as it seeks to embrace economic and social sustainability and emphasizes adding value to the quality of life of individuals and communities. Mak and Peacock [29] include social issues in their definition, such as employment and income and environmental preservation.

The industrial environment faces, in general, the search for innovative technologies, with rationalization in the use of resources eliminating or reducing waste. For Caprian [30], improvement in the production system, or eco-efficiency, and its internal and external integration in the search for the necessary transformations in the current business environment is an accepted concept and refers to strategies that aim to maximize the efficiency of production processes and minimize the negative impact on the environment. Delgado [31] points out that the sustainable production method is the set of actions that implement conscious changes in nature to meet social needs. It also looks to preserve the environment, ensuring the ability to meet the needs of future generations and reducing the wastage of raw materials and inputs.

### 3.1. Environmental Sustainability

To minimize the environmental impacts caused by the construction industry, the paradigm of sustainable construction proposed by Kibert [32–34] emerged. There are authors that understand sustainable construction as the search for technologies and processes that aim to harmonize human interference in nature with construction involving low environmental impact and including recycling processes and the consideration of energy use. Brooks and Rich [35] say that among the goals of sustainability in civil construction is the reduction of energy and water consumption, both from the construction process and throughout the entire useful life of the building. Bohana [36] says that there should be concern regarding reduced emission of pollutants before, during, and after the completion of the work. It is for this reason that sustainability plays a significant role in construction. The sustainable construction method is the set of actions that implement conscious changes in nature aimed at social needs and the preservation of the environment, ensuring the ability to meet the needs of future generations [31].

It is clear that the construction industry is a sector with activities with a high impact on the environment and that the sector needs to adopt some strategic actions to minimize these impacts. The scope of these actions is related to the alignment of the sector with the concept

of sustainable construction [1]. Sousa e Castro [37] defines sustainable constructions as those in which architects and engineers use practices and tools that preserve the environment in construction activities, through proper projects, and in the execution of works.

For Li and Achal [38], sustainability in the construction industry must be addressed not only in projects and works, but also considering the raw materials involved, including their capture, transformation, and the utilization process. That is why it is important to think and plan the entire process for the long term. Design the building to be energy efficient, and seek the proper and rational use of water [39]. Expand the search for environmentally correct materials, as well as achieve comfort and quality in built environments to preserve the environment throughout the life cycle of buildings.

According to ISO 15392:2019 [40], a sustainable building can moderately support or improve quality of life and harmonize with the climate, tradition, culture, and environment in the region in which it is located. At the same time, it conserves energy and natural resources, recycles materials, and reduces hazardous substances within local and global ecosystems throughout the building's lifecycle. Silva [41] defines sustainable construction planning as general principles of sustainable construction: the passive use of natural resources; energy efficiency; water management and savings; waste management in buildings; the quality of the air and the indoor environment; thermoacoustic comfort; the rational use of materials; and sustainable planning of the work.

For sustainable construction that reduces the exploitation of natural resources, Lopes [42] says that civil engineers and architects must think about their projects considering the environmental aspects of the construction site and the sources of raw materials to be used, such as the composition of concrete, for example, which is an input highly demanded by civil construction and which produces a high environmental impact in its production chain.

The environmental impact of civil construction works begins with the subdivision of green areas to make room for the construction of buildings. Sakale [43] says that earthworks and the felling of trees are themselves the cause of enormous environmental impact. D'Adamo [44] remarks that the production of building materials and the consumption of water and electricity during the operation phase are other environmental factors.

*3.2. Economic Sustainability*

Economic sustainability means managing the economy as nature runs its own business. The concept of economic sustainability is based on the importance of balancing human well-being and the well-being of the environment. Phatak [45] emphasizes the need to move away from the conventional economic model, which presupposes unlimited access to natural resources, and to build a new vision in which the management of the use of materials and energy also prioritizes people and other forms of life on the planet. It is important to remember that the planet is an integrated system; for example, plants use carbon dioxide and nutrients to produce oxygen, and animals use this oxygen and create carbon dioxide, so nothing is wasted [46].

Economic sustainability, also known as the circular economy, is the opposite of the linear economy whose motto is "take, do, waste". Aluchna and Rok [47] focus on the implementation of circular economy principles in businesses, highlighting the potential benefits in terms of resource efficiency, reduced environmental impact, cost reduction, and job creation.

It is possible to divide the materials that make up the cycle of the sustainable economy into two main categories: technical materials and biological materials. Technical materials have typical life cycles. They are raw materials extracted from nature, and the product is then manufactured and subsequently transported to be used until the end of its useful life [48]. Often, on this topic, the association between the end of life and the importance of recycling arises, but for the economy to be truly circular, that is, to be sustainable, some things must happen before the end of a product's useful life.

Johnson [49] emphasizes the need for a cohesive management strategy, such as the Sustainability Improvement Program (SEP), to inform industrialists and guide sustainable product design and life cycle management. Sustainability in the life cycle starts with the use of resources that have already been extracted, i.e., already processed. For example, to make a product that has copper among its components, the industry must source copper from processed sources, such as cables or old wires, as this approach is better than extracting copper from the ground. When manufacturing a product, industries can design it so that disassembling the product is easy, meaning the copper in it can quickly be removed for recycling. They can also manufacture products with a longer service life, and there is the possibility of successfully assembling and repairing the same product. The industry also needs to design products whose production involves minimal energy consumption.

When the product has exhausted its useful life, it should be recycled. Recycling is a key point associated with the life cycle. Silva and Gouveia [50] discuss the need for reuse options in product design and the concept of a producer's environmental responsibility to achieve higher levels of recycling and reuse. A necessary question is whether the recycled material keeps its quality when used in similar applications or faces downsizing to a lower-quality material. For example, Oksana [51] talks about the challenges faced by the plastic recycling industry in supporting high-quality recycling, especially for applications in direct contact with consumers. Thus, it is possible to declare that the concept of economic sustainability embeds the obligation to recycle all products, either for other functions or as part of returning to the same production chain so that, in short, nothing goes to waste and everything returns.

The second category of materials in the sustainable economy is biological, i.e., food grown or collected from nature that is processed and transported before reaching the consumer. Once consumed, they can be used as biogas or biochemicals or go through the composting process [52] to finally return to nature and restore ecosystems. Again, it is important to remember that in a sustainable economy, nothing goes to landfill. Two transition strategies can be extremely useful in creating a sustainable economy: substitution and dematerialization. Both substitution and dematerialization can be effective strategies for creating a sustainable economy. Kronenberg [53] concludes that the shift from material to non-material consumption can generate ecological and economic benefits. Substitution refers to the use of different resources to achieve the same goal, and dematerialization, in turn, refers to a decrease in the use of resources to serve the same economic function in society.

Spangenberg [54] advocates for an economy with limited resources, with a limit on the use of resources and alternative allocation mechanisms, because the creation of a truly sustainable economy is only one step towards global sustainability considering other broader aspects to achieve 100% sustainability, such as climate change, the search for sustainable energy, investment in sustainable agriculture, and social sustainability.

Economic sustainability recognizes the value of natural resources and ecological services that sustain life. It recognizes that the economy is part of society and that it is part of the environment. This means that all social and economic progress ultimately depends on the environment [55]. Figure 1 shows how to understand sustainability. The largest circle stands for the environment: the ecosystem services and natural resources necessary for humanity to live and thrive. The middle circle is society or human capital. The economy is the smallest circle, as it is governed, regulated, and structured by the other two circles. The economy depends on the ecosystem and human capital to thrive.

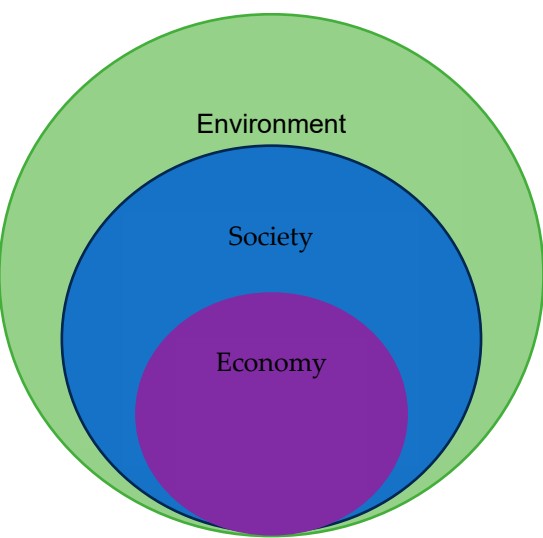

**Figure 1.** Sustainability circles.

### *3.3. Social Sustainability*

The construction industry faces significant challenges in achieving social sustainability, as described by Neto and Farias [56] when talking about the dynamic nature of sustainability in this industry. Franco [57] discusses the potential of Industry 4.0 technologies to increase productivity, efficiency, and safety in construction, which can contribute to social sustainability. Majdalani [58] also highlights the importance of environmentally and economically sound design and development techniques that pursue sustainable construction.

There are different perspectives on social sustainability. Mak and Peacock [29] compared case studies in the UK, US, and Australia to examine the success factors for socially sustainable developments. Dunn [59] discusses the evolution of social sustainability and the challenge of a precise definition of the term. Bijl [60] emphasizes the importance of social aspects in sustainable development, including social capital and community actions. Social sustainability can be defined as the best way to meet human needs within a framework of ecological constraints.

Another theory about human needs was developed by Chilean economist Max-Neef [61]. In this work, the author points out that human needs are finite and can be classified into nine fundamental needs: (a) subsistence, such as food, water, and shelter; (b) protection, because every human being needs a safe place to live with social security; (c) affection, such as contact with friends and romantic relationships; (d) understanding, which involves critical awareness, respect, and tolerance; (e) participation in the community, since participating in decisions that affect society is a human need; (f) idleness, i.e., the need to have some free time to relax and time that leads to learning; (g) creativity, i.e., the capacity for innovation, which leads to problem solving; (h) identity, such as knowing one's place and group of belonging; and (i) freedom, which is being able to make choices of one's own free will. According to Max-Neef [61], these needs are constant in all human cultures and throughout different historical periods. What changes is how these needs are met.

While these needs are the same as those that the Romans and ancient peoples used to have thousands of years ago, some of how they are met today differs in some ways. For example, the need for protection, participation, or freedom is quite different when we think of a society that values free choice, the absence of compulsory labor, and the right to move around in public spaces [62]. While some factors satisfy only one need (for example, security supplies protection), other factors satisfy several needs at the same time, e.g., rehabilitation satisfies the need for protection and identity. These are called synergistic satisfiers. Max-Neef [61] also differentiates other types of factors called destroyers, which are those that supposedly, by satisfying one need, prevent the fulfillment of many others. For example, censorship aims to satisfy society's need to protect itself from harmful infor-

mation. However, at the same time, it impedes attention to the need for understanding, participation, creation, identity, and freedom.

Other factors are pseudo-satisfactory; for example, fashion and trends can generate a false sense of identity, but over time, when fashion passes, there is a loss of identity and belonging. Max-Neef [61] organized the satisfaction needs into four categories: being, having, doing, and interacting.

By working to reduce and eventually drop human contributions that systematically undermine people's ability to meet their needs, there is a movement toward social sustainability [44]. First, by examining the activities, products, and services generated by human action through the lens of associated unsatisfactory, satisfying, pseudo-satisfying, and destructive human needs, this perspective may explain social sustainability differently. Second, when looking for ways to improve or replace an unsustainable practice, the need to step back on the actions taken and seek a distinct perspective may arise.

Thus, questions arise about products and services: Why is this product here? What needs does it satisfy? Can you also satisfy or enhance activities with other products or services? For example, a festival brings together tens of thousands of people to meet their needs for participation, creation, and identity, but is there a way to invent a separate way to satisfy the same needs with fewer carbon emissions and fewer impacts on local ecosystems? Yes, by inventing new ways to satisfy the needs of identities and freedom that do not require buying or consuming as much as possible. When a satisfactory sustainable factor is found, it can be improved by turning it into a synergistic factor [63].

*3.4. Build Sustainable Strategies*

Considering the robust scope involved in the concept of sustainability, a more sustainable building can encompass different strategies, ranging from the specification of materials to the urban scale, involving design, construction, and use, including the following strategies:

1. Energy saving through thermal insulation, high-performance windows, natural lighting, renewable energy generation capabilities, and energy-efficient equipment [5].
2. Thinking as a communitarian, valuing public transport, and easing pedestrian and bicycle traffic [64].
3. Reducing material consumption; optimizing designs to take advantage of small spaces and using materials more efficiently; and reducing waste as well as reducing costs [65].
4. Preserving or restoring the ecosystem and biodiversity. In damaged areas, trying to reintroduce native species. Further, protecting trees and topsoil [27].
5. Choosing low-impact materials and making projects that last and adapt. The longer a construction lasts, the longer the environmental impact period. Designing an adaptable building, especially if it is for commercial purposes [66].
6. Saving water and installing energy-efficient piping and equipment, as well as collecting and using rainwater and separating water from sinks and showers for reuse in garden irrigation [67].
7. Creating a safe and comfortable indoor environment, ensuring the health of its occupants. Allowing daylight to penetrate in as many rooms as possible and supplying continuous ventilation [68].
8. Minimizing construction and demolition waste so that separation and recycling pay off economically [69].
9. Creating environments that focus on the indoor environmental quality of occupants to ensure their health [70].
10. Specifying a risk-based approach tailored to each building, emphasizing heritage aspects. This approach should include constant vigilance, good cleaning practices, and improved fire protection measures [71].
11. Recycling or rehabilitating existing buildings instead of building new spaces [72].

## 4. Building Rehabilitation and Sustainability

Building rehabilitation is a concept in which interventions in a particular building aim to improve the habitability, safety, and energy efficiency of buildings and restore the functionality of a degraded or obsolete building. Interventions include the repair or replacement of structural elements, the renovation of facades and roofs, changes in the interior configuration, or the introduction of new uses for the building. This is an important action for the preservation of built heritage and the promotion of urban rehabilitation.

For the rehabilitation of buildings to be carried out effectively, it is necessary to carry out in situ measurements, monitor the buildings, and interpret the results based on the physical phenomena that cause the pathologies that can be found in the building assessment [72]. Rehabilitation solutions also involve financial aspects related to investments to meet the needs and demands of intervention. It is important to know the construction technologies and their costs to find the best technical and economic balance [73].

Rehabilitation will be an even more important activity, as existing buildings need maintenance and renovation in the future. In Portugal, for example, urban regeneration is a priority, whether it be from an environmental, economic, or social point of view [74]. All interventions must measure and evaluate the properties of materials and the performance of buildings. According to Andersen [75], much of the experimental testing is done in situ, where buildings are monitored to evaluate their in-service behavior.

On the one hand, it is important in rehabilitation works to know the physical phenomena responsible for the pathologies found in the inspections carried out in the buildings. For this, it is important to know the various inspection methodologies and techniques, as well as the tests and measurements necessary to understand the real causes of the pathologies, and then propose interventions to be carried out in the building [76]. Only interpretation of the measurements (tests) allows for a correct diagnosis as to the causes of the problems in the rehabilitation of the structure. Based on this information, engineers can find the best intervention and repair methodology. On the other hand, rehabilitation solutions involve considerable financial aspects in project workflow planning. The ability to recognize technologies and their costs is decisive in achieving the best technical and economic balance to carry out the intervention in a building [77].

Concerning the rehabilitation of buildings in engineering schools in Portugal, it is necessary to reflect on the methodological aspects of rehabilitation, since, according to Appleton [78], the future of civil construction will involve rehabilitation. In the most developed countries in Europe, almost half of the investment in the construction sector is already focused on rehabilitation because they are countries that have an immense built heritage, much of it in concrete, which needs a significant set of works to be maintained, and these works are also required to correctly comply with the new requirements applied to buildings [72]. From the point of view of environmental sustainability, the rehabilitation is mainly sustainable; in the case of concrete structures in Portugal, there is currently a huge dynamic for the maintenance of old buildings in the historic centers [79].

Appleton [80] says that buildings in urban centers will soon need rehabilitation, as the reinforced concrete constructions of the 1960s, 1970s, and 1980s require works that allow for the dignity and functionality of these buildings to be restored. According to the author, all engineering efforts are aware that everything that is built will need maintenance in the future, regardless of its location. There are a variety of areas within engineering where maintenance must occur systematically and periodically. Therefore, rehabilitation needs a lot of study [81].

By applying the building rehabilitation strategy, the works generate less impact on the environment, as they consume less water and energy resources and reduce the generation of waste; during the intervention process, the choice of modern and less polluting materials can reduce the environmental impacts of civil construction, as long as those involved in the planning use the most up-to-date construction techniques [82].

To Grijalba-Aseguinolaza and Eizaguirre-Iribar [83], rehabilitation is also an action with a huge social impact. In addition to the preservation and maintenance of historical

heritage, rehabilitation also contributes to the transfer of history to future generations. The recovery of buildings in previously degraded urban regions generates a new life cycle for these regions. Rehabilitation can also be a unique real estate occasion, as a rehabilitation operation based solely on energy efficiency may not be extraordinarily interesting for a developer. Rehabilitations should add more value to upgraded buildings. This is because old buildings were designed with old plans and adapted to a non-existent reality [84], and costs for developers increase when it is necessary to change and update the plans of old buildings. This process often involves modifying the structure of the building, and this is not simple. To change the structure, it is necessary to have specific and proper technology for each intervention [85]. In this way, rehabilitation is also a technological innovation. For example, applying a technology that allows you to change the internal structure of a building while maintaining the original façade at a low cost in the overall structure of the building has a positive impact on the region surrounding the building, as there is no change in the esthetics of the facades. There is also the addition of new responsibilities for construction, respecting and adapting the innovative design to the new standards and regulations for buildings that involve both energy efficiency and other environmental issues, such as water consumption and noise emissions. Therefore, technology makes rehabilitation interventions interesting for developers [76].

Some existing buildings are not of great historical importance. Therefore, rehabilitation for them does not serve the interests of historical heritage, but rather sustainability issues. They avoid demolition and expenses by reusing natural resources by conducting interventions that will provide a longer useful life for buildings [86]. In building rehabilitation projects, the choices are made using new parts and new materials, which are added to old parts and materials. Sobotka [87] discusses the need to see compatibilities as well as quality requirements integrated with costs in the proposed intervention.

Considering that the rehabilitation of buildings is any intervention conducted, essentially, within the urban space, a confusion between concepts arises, as some consider that urban rehabilitation is only operating in a building within the city when the concepts of urban rehabilitation are broader. The rehabilitation is acting mainly in the space of the city, in the urban space, and includes dealing with the public space, including infrastructure [88]. When talking about urban rehabilitation, it often refers only to the renovation of buildings. However, for a real understanding of the rehabilitation process, it is essential to re-establish a balance between the original identity of the building and the transformations resulting from the proposed intervention [89]. The result of this balance between the original identity of the structure and the transformations generated by the rehabilitation makes many people, especially technical professionals, hesitant to get involved in the process. This is due to the complexity of the work, which requires a deep knowledge of the building that is the object of the intervention. However, this same need, although challenging, is what provides a certain charm to rehabilitation. It is through this challenge that engineers and architects are faced with different realities throughout the entire process, from design to completion [90].

## 5. Final Considerations

Global construction, which accounts for 39% of $CO_2$ emissions, faces many environmental challenges. Countries, including Brazil and Portugal, are looking for sustainable methods and regulations for efficient construction and pollution reduction. The concept of sustainability appeared in 1983 and was expanded in 1992 at the ECO 92 UN conference encompassing environmental preservation and quality of life. Chakravarty highlights the economic and social scope of sustainable development. The construction industry seeks eco-efficiency and sustainable production with the aim of meeting social needs and preserving the environment and looks to harmonize human interventions with low environmental impact, including reductions in energy and water consumption and pollutant emissions. The rehabilitation of buildings aims to reduce environmental impact and prioritize sustainability over demolition, promoting the preservation of resources. Economic sustainability seeks to balance human and environmental well-being, moving away from the conven-

tional model. In construction, the challenges for social sustainability involve Industry 4.0 technologies and divergent perspectives on fundamental human needs. Rehabilitation aims to improve buildings to preserve heritage, and the intervention process requires technical and methodological ability, physical understanding, and financial consideration for lasting success. The review of the literature on the sustainability of building rehabilitation reveals that, although associated with civil engineering and architecture, rehabilitation has essential elements of sustainability. The texts listed and consulted show the connection between the processes of building rehabilitation and concerns with sustainability in its most varied dimensions, whether environmental, economic, or social. With this work, we hope to highlight the importance of continuing studies on sustainability in rehabilitation works to strengthen the field of building rehabilitation with subsidies about its positive impacts that go beyond the renovation or retrofit of buildings.

**Author Contributions:** Conceptualization, A.M.C.P.d.O., J.C.G.L. and A.P.K.; methodology, A.M.C.P.d.O., J.C.G.L. and A.P.K.; validation, A.M.C.P.d.O., J.C.G.L. and A.P.K.; formal analysis, A.M.C.P.d.O.; investigation, A.M.C.P.d.O.; writing—original draft preparation, A.M.C.P.d.O.; writing—review and editing, A.M.C.P.d.O., J.C.G.L. and A.P.K.; visualization, A.M.C.P.d.O., J.C.G.L. and A.P.K.; supervision, J.C.G.L. and A.P.K.; funding acquisition, J.C.G.L.. All authors have read and agreed to the published version of the manuscript.

**Funding:** This work was supported with Portuguese national funds by FCT—Foundation for Science and Technology, I.P., within the project C-MADE-UIDB/04082/2020.

**Institutional Review Board Statement:** Not applicable.

**Informed Consent Statement:** Not applicable.

**Data Availability Statement:** The raw data supporting the conclusions of this article will be made available by the authors on request.

**Conflicts of Interest:** The authors declare the non-existence of conflicts of interest.

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
