# Peer review of "Building Rehabilitation: A Sustainable Strategy for the Preservation of the Built Environment"

_sustainability, doi:10.3390/su16020553_

Round 1
Reviewer 1 Report
Comments and Suggestions for Authors
One of the main questions addressed in the research is that ‘building rehabilitation and sustainability must walk together to ensure the preservation of the built environment and environmentally conscious practices’. Additionally, it is emphasized that ‘civil construction is one of the most polluting industries with a high carbon footprint impact’, and that the ‘building rehabilitation emerges as an effective strategy to reduce this impact, promoting the reuse of materials and more efficient technologies’.
In synthesis, the article ‘aims to identify the predominant themes studied in this context’. According to the authors, the study ‘focuses on the existing buildings rehabilitation as a sustainable strategy and presents the quantitative profile of academic publications over the last 10 years, identifying the main themes studied’. While this aspect is a strong point of the work, it is underexplored in the content.
Despite possessing moderate originality, the topic remains relevant within the field, but does not offering prominent contributions. To enhance the uniqueness of the work, a more thorough exploration of new findings based on a review of articles in the Scopus repository is essential. This aspect could significantly distinguish the text from other published materials in the subject area, providing a more substantial and novel results.
The improvement of the methodological approach for content analysis of database-searched articles is crucial and should involve specific controls and quality checks. It is advisable to provide detailed descriptions not only regarding the frameworks utilized but also in relation to sustainability aspects, particularly those highlighted in item 2, ‘The Concept of Sustainability in Civil Construction’. This would insure internal scientific alignment, organizes the work comprehensively, and enhances its scientific robustness. Furthermore, proposing and testing an investigative hypothesis based on a solid experimental design would contribute significantly, not only strengthening the study's scientific foundation but also increasing its reproducibility.
While the conclusions align consistently with the presented evidence and arguments, they currently adopt a synthesis format and could benefit from a more analytical perspective. Strengthening the final considerations with support from recent scientific findings would enhance the overall depth and analytical richness of the work. This approach would contribute to a more nuanced and comprehensive exploration of the main question posed in the study.
The references are suitable, yet the bibliometric results are underexplored. Could be consider the incorporation of additional dimensions such as geographic-locational, technical-technological, and legal-institutional, among to those mentioned earlier. However, it is worth highlighting the contemporary nature of the sources, with over 80% from the last decade and nearly 55% from the last five years. This temporal focus enhances the relevance and currency of the study.
Depending on the detail of the analytical content of the articles found in the bibliometric search, new figures and tables could be added. For the latter existing ones, it is recommended to also add quantities to improve data interpretation. In general, moderate editing of English language is required for clarity of the text and accuracy of the information, contributing to a more effective communication of the work.
Comments on the Quality of English LanguageSee comments above
Author Response
I thank you for your considerations and I believe that the changes I made meet your suggestions and improve the text.

Reviewer 2 Report
Comments and Suggestions for Authors
First of all, I would like to congratulate the authors for this rigorous and interesting article and for their approach to this field.
Despite being especially focused on Portugal and Brazil, the methodology can be applied to other case studies. The content is very comprehensive and the topic is relevant at the present time. The graphics are few, but sufficient and coherent with the text which in turn is clear and precise.
The references and bibliography are up-to-date as befits the subject at hand.
I can only say that it is a great contribution to the Journal
Author Response
Thank you for your considerations

Reviewer 3 Report
Comments and Suggestions for Authors
The work carried out by the authors of the review is a very relevant area, since energy saving, environmental safety and restoration of existing buildings is a problem of the international community. However, to improve the quality of the study, the reviewer suggests making significant improvements.
Design notes:
1. Table - 1 The vertical scale must be specified in the diagram
2. Table – 2 in the diagram in the materials, Technology, Water and Sustainability sections, add a name that is not indicated in the figure
3. In section 5 of Final Considerations, there is no need to specify links, I suggest removing the numbering of the link [12].
4. Lines 433-435 indicate 45 references, table 1 also shows publications from 2013-2022, and which publications should be referenced???, a mandatory requirement
5. Table 2 shows the publications of 2015, 2017, 2019, 2020, and which publications should be referenced???, a mandatory requirement
6. Section 4 of the article is more similar to the research methodology, I recommend that you finalize it and put it after the introduction section.
7. In general, I recommend that the authors review the structure of the article, since in this form it looks scattered
A note on the study:
1. Become written with poor statistical indicators. It is necessary to provide statistical and quantitative indicators. For example: the application of a particular strategy
2. The contribution of the authors to the work is unclear, it needs to be supplemented.
3. The reviewer is strongly recommended to supplement the review from different databases (Web of science, Online library Wiley, Science direct, Sage, ASCE library,...), and not only from the Scopus database.
4. Section 5 of Final Considerations needs to be redone taking into account quantitative indicators
5. The abstract of the work needs to be improved, it is written very poorly
6. In the introduction of the article, it is necessary to specify the purpose of the work
7. I also suggest that the authors familiarize themselves with the following works, where the strategy was also considered and designs for energy saving in construction were developed:
https://doi.org/10.1016/j.egypro.2017.07.384
https://doi.org/10.15866/irece.v13i2.20933
Author Response

(The authors gave the same response as above.)

Round 2
Reviewer 3 Report
Comments and Suggestions for Authors
The comments are accepted.
I wish the authors further scientific achievements.
Kind regards, Reviewer!!!